# Should We Fear the Frail? A Review on the Impact of Frailty on Liver Surgery

**DOI:** 10.3390/medsci13040253

**Published:** 2025-10-31

**Authors:** Sorinel Lunca, Stefan Morarasu, Raluca Zaharia, Ana Maria Musina, Wee Liam Ong, Gabriel Mihail Dimofte, Cristian Ene Roata

**Affiliations:** 1Grigore T. Popa University of Medicine and Pharmacy Iasi, 700115 Iași, Romania; sdlunca@yahoo.com (S.L.); raluca.zaharia11@yahoo.com (R.Z.); musina.anamaria@gmail.com (A.M.M.); william05021990@gmail.com (W.L.O.); gdimofte@gmail.com (G.M.D.); roatacristianene@gmail.com (C.E.R.); 22nd Department of Surgical Oncology, Regional Institute of Oncology (IRO), 700483 Iași, Romania

**Keywords:** frailty, liver cancer, liver surgery, geriatrics, hepatectomy

## Abstract

Background: Frailty is a multidimensional syndrome characterized by reduced physiological reserve and resilience and has become a crucial predictor of outcomes in liver surgery. Unlike chronological age, frailty reflects broader vulnerabilities that significantly influence postoperative recovery. Aim: To review and synthesize current evidence on the relationship between frailty and postoperative outcomes following liver resection, with an emphasis on short-term complications, mortality, and long-term survival. Methods: A comprehensive literature review was performed, drawing on recent meta-analyses, large-scale cohort studies, and prospective observational data. Frailty was evaluated using a range of assessment tools, including the Modified Frailty Index (mFI), Clinical Frailty Scale (CFS), Kihon Checklist (KCL), and claims-based measures such as the Johns Hopkins Frailty Indicator. Results: Across studies, frailty has been consistently linked to a higher incidence of postoperative complications, such as post-hepatectomy liver failure (PHLF), infections, extended hospital stays, and increased mortality. In patients undergoing liver resection for cancer, frailty is also associated with poorer long-term survival. Importantly, frailty serves as an independent risk factor, even after adjusting for age, comorbid conditions, and tumor characteristics. Preoperative identification of frailty enhances risk stratification, informs surgical planning, potentially favoring parenchymal-sparing or minimally invasive approaches, and highlights patients who may benefit from prehabilitation. Conclusions: Frailty is a strong and independent predictor of poor outcomes after liver resection. Incorporating frailty assessment into routine preoperative evaluation can improve surgical decision-making, facilitate informed patient counseling, and optimize perioperative care strategies.

## 1. Introduction

Global life expectancy has shown a consistent upward trajectory and is projected to reach an average of 77.3 years by 2050. This demographic shift imposes increasing demands on surgical services, as a growing proportion of elderly patients will become candidates for major surgery. Consequently, postoperative outcomes in this population must be carefully evaluated, given that approximately one in seven older adults die within the first year following major surgery. When considering major liver surgery which is one of the most complex abdominal interventions, greater caution is required when performing major resections in high-risk elderly patients. Selective prehabilitation of high-risk patients undergoing major hepatectomy is usually performed guided by traditional risk assessment models which often fail to capture the full spectrum of vulnerabilities associated with aging. To address this, frailty has emerged as a potential quantifiable syndrome capable of reflecting one’s ability to handle surgical stressors. Unlike chronological age, which is an inadequate standalone predictor of surgical risk, frailty encompasses a range of factors, including sarcopenia, cognitive decline, and multimorbidity. These components should collectively contribute to worse surgical outcomes, prolonged hospital stays, and higher readmission rates [1,2,3,4,5,6].

When it comes to primary and secondary liver cancer, surgical resection remains the gold standard treatment for patients deemed operable, offering the best long-term oncologic outcomes when sufficient hepatic reserve is preserved. However, in patients who are not suitable surgical candidates due to comorbidities, limited functional reserve, or unfavorable anatomy, locoregional therapies such as thermal ablation or transarterial chemoembolization, serve as first-line option [7,8]. In poor surgical candidates who have an indication for ablation, the overall survival (OS) is noninferior to that of operated ones [9,10]. This highlights the importance of stratifying patients based on their surgical risk as in many cases alternative treatments may offer similar oncological outcomes with superior morbidity and short-term mortality. Quantifying frailty may offer a better understanding of patient’s physiological reserve and could be used as a clear unbiased indicator of patient inoperability. Herein, we aim to review and synthesize current evidence on the relationship between frailty and postoperative outcomes following liver resection, with an emphasis on short-term complications, mortality, and long-term survival.

## 2. Methods

A comprehensive literature search was performed across PubMed/MEDLINE, Embase, Scopus, Web of Science, and the Cochrane Library from the earliest date to 10 August 2025. The search combined keywords and MeSH terms such as “*frailty*,” “*frail elderly*,” “*sarcopenia*,” “*hepatectomy*,” “*liver resection*,” “*liver surgery*,” “*morbidity*,” “*mortality*,” “*outcomes*,” “*complications*,” “*readmission*,” and “*liver failure*.” Boolean operators (e.g., *frailty AND liver resection AND outcomes*) were applied to refine the results.

## 3. How Is Frailty Measured?

In everyday clinical practice, frailty is often recognized instinctively—clinicians may identify a frail patient the moment they walk into the room. An elderly, undernourished individual with visible mobility issues is typically perceived as frail. When such a patient also presents with multiple comorbidities and functional or nutritional impairments, many surgeons may hesitate to offer major surgery or, at the very least, take a more cautious and realistic approach to discussing recovery. This intuitive recognition of frailty, however, is based largely on subjective experience. It was not until frailty was formally conceptualized as a clinical syndrome that efforts began to define and measure it objectively in high-risk patients [11]. Two principal models of frailty have since been described in the literature: Fried’s frailty phenotype and the deficit accumulation model.

Fried’s phenotype model characterizes frailty through a physical lens, using five measurable criteria: slowed gait speed, muscle weakness (often assessed via grip strength), unintentional weight loss, self-reported exhaustion, and low levels of physical activity. This model captures the physical vulnerability often seen in older adults, including those without clear chronic illnesses. However, it also overlaps with aging-related chronic diseases, though it is not limited to them.

In contrast, the deficit accumulation model, commonly operationalized as the Frailty Index, views frailty as the cumulative effect of various health deficits. These may include symptoms, signs, chronic conditions, functional impairments, and laboratory abnormalities. This model provides a broader and more quantitative picture of vulnerability, making it the most widely used approach in frailty research and clinical assessments [12,13,14,15].

In liver surgery, multiple frailty assessment tools have been employed, each with its own methodology, advantages, and limitations. Choosing the right tool and understanding the strengths and weaknesses of each is critical for interpreting the evidence and applying frailty assessment effectively in clinical decision-making. Figure 1 depicts the most widely used frailty scales in liver surgery research.

The Fried Frailty Index (FFI) is one of the most extensively validated and widely used tools for assessing physical frailty in older adults. It conceptualizes frailty as a clinical syndrome, defined by the presence of at least three out of five criteria: unintentional weight loss, self-reported exhaustion, muscle weakness (typically assessed via handgrip strength), slow walking speed, and low physical activity. The FFI has demonstrated strong predictive value for adverse outcomes such as disability, falls, hospitalization, and mortality. Its key strengths include its simplicity, objectivity, and clear focus on physical function, making it well-suited for both clinical evaluation and intervention planning. However, limitations include its exclusion of cognitive, psychological, and social domains, partial reliance on self-reported information, and the requirement for specific equipment (e.g., a dynamometer) to assess grip strength [12,13,14,15].

The Modified Frailty Index (mFI) is among the most employed frailty assessment tools in surgical research, particularly in studies based on the American College of Surgeons National Surgical Quality Improvement Program (NSQIP). The traditional 11-item mFI includes variables such as diabetes, congestive heart failure, and functional dependence, while a more concise 5-item version has also been validated. Both versions have shown strong predictive performance for postoperative morbidity and mortality in patients undergoing hepatectomy [16,17]. The mFI’s primary strengths lie in its objective, retrospective applicability, making it highly suitable for large-scale database analyses. Nevertheless, it does not account for physical performance or cognitive function, limiting its comprehensiveness in certain clinical scenarios.

The Clinical Frailty Scale (CFS) is a visually anchored, 9-point tool based on clinician judgment, integrating physical function, comorbidities, and degree of independence. Initially developed by Rockwood et al. [18], the CFS has been validated in multiple liver surgery populations, including elderly patients undergoing resection for hepatocellular carcinoma (HCC) and colorectal liver metastases (CRLM), where it has consistently predicted major complications and prolonged hospitalization [19,20]. The scale’s intuitive format and ease of use make it highly suitable for bedside assessment, although its subjective nature and potential for inter-rater variability remain important limitations.

The Kihon Checklist (KCL) is a 25-item geriatric screening questionnaire developed in Japan, encompassing domains such as physical function, cognition, nutrition, mood, and social engagement. It has been applied in several prospective studies involving elderly patients undergoing liver resection [21,22]. The KCL provides a multidimensional profile of patient vulnerability and has demonstrated effectiveness in predicting postoperative functional decline. However, its use remains largely restricted to East Asian populations due to linguistic and cultural specificity.

The Adjusted Clinical Groups (ACG) frailty indicator is embedded within the Johns Hopkins ACG System, a population health analytics platform. This indicator identifies frailty based on the presence of one or more clusters such as malnutrition, dementia, incontinence, decubitus ulcers, falls, unintentional weight loss, severe visual impairment, mobility issues, poverty, or barriers to care. The ACG frailty indicator has recently been applied to hepatectomy cohorts using large administrative datasets such as the National Readmissions Database (NRD), demonstrating its predictive value for complications (e.g., infections, sepsis, venous thromboembolism), increased length of stay, discharge to rehabilitation facilities, and in-hospital mortality [23].

Other frailty assessments, such as the Geriatric-8 (G8) screening tool and gait speed tests, have been investigated in smaller cohorts. The G8, designed specifically for oncogeriatric populations, is effective in identifying older adults at risk for poor treatment tolerance [24]. Meanwhile, gait speed, particularly thresholds below 1.0–1.1 m/s, has emerged as a strong, objective predictor of overall survival in older adults [25]. Although these tools offer reproducible, performance-based metrics, their use is generally limited to prospective settings and requires time, space, and equipment.

In summary, no single frailty assessment tool is universally optimal. However, the mFI and CFS remain among the most widely utilized and clinically practical instruments in the context of liver surgery. Objective measures such as gait speed provide valuable prognostic information but are less commonly implemented. Ultimately, the choice of frailty tool should reflect the clinical context, study design, and data availability (Table 1). Standardization of frailty assessment is encouraged in future prospective trials in liver surgery.

## 4. Prevalence of Frailty in Hepatectomy Patients

The reported prevalence of frailty among patients undergoing hepatectomy varies widely, ranging from 15% to 35% across different studies and populations [26,27,28]. A 2024 meta-analysis by Zhang et al., which included 128,868 patients who underwent hepatectomy, estimated the pooled prevalence of frailty at 23% (95% CI: 17–28%) [27]. Similarly, Lv et al. analyzed a cohort of 84,096 patients undergoing liver resection for either primary hepatic malignancy or CRLM and found that 28% of patients met frailty criteria [28]. Two recent meta-analyses encompassing studies conducted between 2016 and 2023, reported frailty prevalence rates of 24.1% (10 studies, 71,102 patients) and 35% (18 studies, 38,157 patients) among patients with liver cancer [29,30]. Frailty was more commonly observed in individuals with primary liver cancer, with reported rates of 29% in HCC and 27% in intrahepatic cholangiocarcinoma (ICC), compared to 18% in those with CRLM. Notably, among patients undergoing hepatectomy for primary liver malignancies, those with HCC were significantly more likely to be frail than those with ICC (41% vs. 20%, respectively).

Most of the published data on frailty in hepatectomy patients originate from studies conducted in the United States and Japan, with relatively fewer contributions from European institutions. Regional variation in frailty prevalence is evident. Among patients undergoing hepatectomy, the prevalence of frailty was 26% in East Asian countries and 18% in Europe and North America. In liver cancer populations specifically, frailty was most prevalent in Asia (40%), followed by Europe (38%), and North America (25%). These differences may reflect variations in patient demographics, clinical practices, frailty assessment tools, and access to prehabilitation or geriatric services [27,28,29,30].

## 5. Postoperative Morbidity, Mortality, and Length of Stay

Frail patients undergoing liver resection consistently face significantly increased risks of postoperative morbidity and mortality. Two recent meta-analyses underscore the strong association between frailty and adverse short-term surgical outcomes [28,29]. In a comprehensive 2024 systematic review of 13 studies encompassing over 84,000 patients, frailty was associated with a 1.69-fold increased risk of any postoperative complication and a 2.69-fold higher likelihood of major complications (Clavien–Dindo grade ≥ III) compared to non-frail patients. Furthermore, frail individuals exhibited a markedly elevated risk of early postoperative mortality, with 30-day mortality 4.6 times higher and 90-day mortality 2.5 times higher than in non-frail counterparts [28]. Another 2024 meta-analysis focused specifically on patients undergoing oncologic liver resections found that frailty was linked to a 2.9-fold increase in overall morbidity, a 2.2-fold higher rate of major complications, and a 1.3-fold higher incidence of PHLF. This analysis also noted increased 30-day readmission rates (1.3-fold) and significantly elevated short-term mortality in frail patients (*p* < 0.001) [29].

These meta-analytic findings are corroborated by multiple large-scale cohort studies. For instance, Louwers et al., analyzing 10,300 liver resections from the NSQIP database, demonstrated that higher frailty scores based on the 11-item mFI were independently associated with increased odds of severe complications (Clavien–Dindo grade IV), elevated 30-day mortality, and prolonged hospital stays. Notably, frailty predicted poor outcomes regardless of whether patients underwent minor or major resections [31]. Similarly, a study by Chen et al. found that patients with higher mFI scores undergoing combined colorectal and liver resections had significantly higher rates of both overall and serious complications, along with extended hospital stays. In multivariate models, frailty independently predicted morbidity, while age and hepatectomy type did not [32]. Gani et al. expanded on the concept by developing a revised frailty index incorporating physiological parameters such as serum albumin, hematocrit, American Society of Anesthesiologists class, body mass index, and extent of resection. This updated index demonstrated superior predictive value for postoperative complications, length of stay (LOS), and mortality, while the traditional mFI remained predictive of complications but was less reliable for mortality and LOS outcomes [33]. These findings suggest that integrating objective clinical and laboratory data into frailty assessments may enhance predictive accuracy.

Beyond general morbidity, frail patients are prone to specific types of complications at disproportionately high rates. Shahrestani et al., using the JHACG frailty indicator in a national cohort of patients with colorectal liver metastases, reported significantly higher rates of infectious complications (pneumonia, sepsis), wound dehiscence, urinary tract infections, postoperative anemia, and thromboembolic events among frail individuals. These patients also experienced longer hospital stays, increased likelihood of non-home discharge, higher readmission rates, and greater in-hospital mortality [23]. These trends reflect the vulnerability of frail individuals, who often suffer from diminished physiological reserve and immune dysfunction, predisposing them to systemic complications [28]. Several studies have further highlighted the elevated risk of PHLF and multi-organ dysfunction necessitating ICU care among frail patients [34,35].

Length of hospital stay is consistently longer in frail populations. According to Lv et al., frail patients remained hospitalized for an average of 3.7 days longer following hepatectomy (95% CI: 1.45–5.85 days) [28]. Supporting this, Yamada et al. reported prolonged postoperative stays in frail octogenarians undergoing liver surgery in a Japanese center [36]. The extended LOS in these patients is likely a result of both higher complication rates and delayed functional recovery [21]. Importantly, frailty also predicts postoperative loss of independence. Shahrestani et al. observed a strong association between frailty and discharge to skilled nursing or rehabilitation facilities rather than to home [23]. However, not all studies reported uniformly adverse outcomes associated with frailty. A prospective multicenter study in Japan utilizing the KCL frailty tool found no significant differences between frail and non-frail patients in terms of 30-day mortality, overall complications, major complications, or median LOS. Interestingly, 90-day mortality was significantly higher among frail patients, suggesting that while some may survive the immediate postoperative period, they remain vulnerable to later decline and death [21]. The discrepancies between the findings of Tanaka et al. and previous studies examining the impact of frailty on outcomes after liver resection can be explained by several factors. First, their study focused exclusively on independent elderly patients deemed suitable for surgery, representing a highly selected, low-risk cohort with limited variability in frailty status. This selection bias likely reduced the ability to detect significant differences between frail and non-frail individuals. Second, the heterogeneity of frailty assessment methods contributes to inconsistent results across studies. Whereas earlier research often employed the modified Frailty Index (mFI)—a comorbidity-based tool that may not fully capture physiological vulnerability—Tanaka et al. used a clinical assessment approach, reflecting different dimensions of frailty. Third, ongoing advancements in surgical techniques, perioperative optimization, and enhanced recovery protocols may have mitigated the short-term impact of frailty on morbidity and mortality. Finally, frailty represents a continuum, and in carefully optimized surgical candidates, the threshold beyond which frailty affects outcomes may not be reached. Collectively, these factors suggest that while frailty remains an important predictor of surgical resilience, its influence may be less evident in well-selected, low-risk populations undergoing modern hepatic surgery.

Similarly, a case–control study involving 230 patients showed that frailty (mFI-5) was associated with significantly higher PHLF rates and longer ICU stays in those undergoing major resections, while frailty did not appear to influence complication rates in patients undergoing minor resections. In that cohort, overall morbidity and 30-day mortality were comparable between groups, likely due to careful surgical selection and limited resections in frail individuals [35].

Moreover, a recent meta-analysis encompassing nine studies and 46,949 patients found no significant difference in postoperative LOS between frail and non-frail patients [29]. These findings highlight that the effect of frailty on postoperative outcomes may be attenuated by limiting the surgical extent or through stringent preoperative selection. Nonetheless, one striking observation remains: frail patients with an mFI > 2 undergoing major liver resections experienced an eightfold increase in PHLF incidence compared to their non-frail peers [35].

In conclusion, while variability exists among studies, the overwhelming body of evidence indicates that frailty is a potent predictor of adverse short-term outcomes after liver resection, including increased complications, mortality, and prolonged hospitalization. Table 2 provides a summary of key studies evaluating frailty and postoperative outcomes in liver surgery.

## 6. Long-Term Survival Outcomes

Beyond the immediate postoperative period, frailty significantly impacts long-term survival following liver resection. A meta-analysis by Lv et al. demonstrated that frail patients had markedly poorer long-term outcomes, with a pooled hazard ratio (HR) of 2.89 (95% CI: 1.84–4.53) for overall survival compared to non-frail individuals. This nearly threefold increased risk of long-term mortality remained significant even after excluding perioperative deaths, highlighting the prognostic relevance of frailty beyond the first 90 days after surgery [28].

Several individual studies support the conclusion that frailty independently predicts reduced long-term survival. Okada et al. found that frail patients with HCC had a 5-year OS of 42.7%, significantly lower than the 77.2% seen in non-frail patients (*p* = 0.005). On multivariate analysis, frailty remained an independent predictor of poorer OS, even after adjusting for tumor characteristics and comorbidities. Notably, frail patients had a higher rate of extrahepatic recurrence (30.8% vs. 3.6%, *p* = 0.028) and were less likely to undergo repeat curative treatments, suggesting that frailty may limit both physiological resilience and access to additional cancer therapies [22]. In a large multicenter study of 240 octogenarians undergoing hepatectomy for HCC, preoperative frailty assessed by the CFS was strongly associated with both increased 30-day morbidity (OR: 2.060) and reduced long-term survival. Frailty independently predicted worse outcomes across multiple endpoints, including OS (HR: 2.384), recurrence-free survival (RFS; HR: 2.190), and cancer-specific survival (CSS; HR: 2.203). Interestingly, age alone was not an independent predictor of long-term survival [39]. Similarly, Tokuda et al. observed that frailty in patients with colorectal liver metastases was associated with significantly lower 3-year OS (63.9% vs. 89.1%) and CSS (69.3% vs. 91.0%) after hepatectomy. While disease-free survival (DFS) was also lower in frail patients (22.7% vs. 38.6%), the difference was not statistically significant (*p* = 0.054). On multivariate analysis, frailty was the only independent predictor of CSS (*p* = 0.0477) [20].

The mechanism underlying these findings is likely multifactorial. Frail patients may be more susceptible to postoperative complications, functional decline, and decreased physiological reserve, limiting their ability to tolerate cancer recurrence or additional therapies. Moreover, although recurrence-free survival (RFS) is not always significantly affected by frailty, as shown by Okada et al., who reported no statistical difference in recurrence rates between groups, frail patients were less likely to undergo salvage treatments, leading to worse overall outcomes [22]. The finding that frailty affects overall survival but not recurrence-free survival after hepatectomy provides an important perspective on its broader significance in surgical oncology. This pattern suggests that frailty does not directly influence tumor recurrence but instead reflects diminished physiological reserve and resilience, which limit patients’ ability to tolerate postoperative stress, adjuvant therapies, or salvage treatments after recurrence. Frail patients often experience delayed recovery, sarcopenia, or functional decline, making them less likely to be eligible for repeat hepatectomy, ablation, or systemic therapy, thereby compromising long-term survival despite similar oncologic control. Consequently, frailty should be viewed not merely as a surgical risk factor but as a longitudinal marker of physiological vulnerability that shapes a patient’s entire cancer journey. This insight carries important implications for preoperative counseling and shared decision-making, emphasizing that discussions of operative risk should extend to expectations about postoperative recovery, treatment feasibility for recurrence, and quality of life. Moreover, it underscores the potential value of targeted prehabilitation and rehabilitation strategies aimed at preserving functional reserve and improving access to future therapeutic opportunities.

In summary, frailty appears to affect cancer prognosis not primarily through tumor biology, but rather via non-cancer determinants of survival such as comorbidity burden, functional decline, and limited treatment options for recurrent disease. Table 3 provides a summary of key studies evaluating the relationship between frailty and long-term outcomes in liver malignancies.

## 7. Independent Predictive Value of Frailty and Clinical Implications

A consistent finding across the literature is that frailty remains a strong and independent predictor of adverse outcomes, even after adjusting for traditional risk factors. In numerous studies, frailty retains its prognostic significance while variables like chronological age frequently lose predictive value [19,22,32]. These results emphasize that frailty captures a dimension of vulnerability not fully reflected by age, comorbidity indices, or standard laboratory data. As highlighted by Lv et al., frailty likely encompasses subtle physiological impairments, such as sarcopenia, malnutrition, and cognitive decline that directly influence surgical tolerance and recovery capacity [28].

Incorporating frailty assessment into preoperative evaluations enhances risk stratification. For instance, Maegawa et al. showed that combining the mFI with the ALBI grade significantly improved prediction of 30-day mortality and major complications after liver surgery [34]. Similarly, Shahrestani et al. found that predictive models including both frailty and age more accurately forecasted key outcomes such as discharge to care facilities, venous thromboembolism, and postoperative infections, compared to age-based models alone [23].

These findings have important clinical implications. As concluded in our previous meta-analysis frailty represents a “solid predictive risk factor” and should be considered during surgical decision-making, especially in high-risk or borderline surgical candidates [35]. This narrative review builds upon that statement by focusing specifically on frailty tools, scale validation, and cross-study outcome synthesis, thereby extending the scope rather than repeating the earlier analysis. Frailty assessment may guide clinicians toward less aggressive interventions, such as limited resection or even non-operative management, when appropriate. When surgery is pursued, targeted prehabilitation, including physical conditioning and nutritional optimization, may help bolster physiological reserves and improve surgical resilience [34,36]. Moreover, minimally invasive surgical techniques may offer benefits for frail patients by reducing perioperative stress and facilitating faster recovery [34]. Postoperatively, frail individuals should receive enhanced monitoring, early mobilization, and structured discharge planning, including referrals for rehabilitation when indicated. These measures can help mitigate prolonged recovery, prevent functional deterioration, and support a safer transition from hospital to home [23]. Regional variations in perioperative care likely influence how frailty affects outcomes following hepatectomy. In Japan, prehabilitation typically focuses on nutritional optimization such as supplementation with branched-chain amino acids in patients with cirrhosis, along with meticulous risk assessment and prolonged in-hospital recovery supported by intensive nursing care. These practices may help mitigate short-term complications in frail patients but often contribute to longer hospital stays [41,42]. In Europe, ERAS Society protocols are widely standardized and audited, with common features including preoperative cardiopulmonary exercise testing (in select centers), early mobilization, opioid-sparing analgesia, and structured discharge planning. Such measures can reduce surgical stress and accelerate recovery, even in vulnerable patients, though adherence and resources vary between countries and institutions [43,44]. In the United States, ERAS implementation is broad but heterogeneous; while perioperative anesthesia and mobilization practices are well developed, access to formal prehabilitation, home health, and post-acute care services varies considerably, and insurance factors may limit the intensity and continuity of rehabilitation, potentially widening outcome disparities among frail patients [45,46]. Social support systems further contribute to these regional differences: stronger family caregiving traditions in parts of Asia and organized community or home-based care models in parts of Europe contrast with more fragmented support structures in some U.S. settings. Overall, differences in prehabilitation availability, ERAS compliance, anesthesia and ICU management, transfusion and analgesia strategies, and post-discharge support may account for much of the observed regional heterogeneity in frailty-related surgical outcomes. Future multicenter studies should therefore consider region-specific care pathways and report adherence to ERAS and prehabilitation protocols to allow meaningful comparisons. In conclusion, frailty is robustly and independently associated with both short- and long-term adverse outcomes following liver resection. This association holds true across various definitions of frailty and remains significant after adjusting for age, tumor characteristics, and comorbidities [22,28]. As such, routine preoperative frailty assessment should become a standard part of surgical planning. Doing so enables personalized surgical strategies, optimizes perioperative management, and ultimately improves outcomes in this vulnerable patient population.

### 7.1. Frailty in Specific Patient Populations Undergoing Liver Resection

The prognostic significance of frailty becomes especially pronounced within specific subgroups undergoing liver resection—namely, the elderly, patients with primary or metastatic liver malignancies, and those undergoing major versus minor hepatic resections. Understanding how frailty influences outcomes in these distinct populations is crucial for effective risk stratification and individualized surgical planning.

Older adults (≥70 years) constitute a growing proportion of patients undergoing liver surgery. However, chronological age alone is a poor predictor of perioperative risk, and frailty has emerged as a more accurate marker of vulnerability in this group. Yamada et al. reported that frail octogenarians experienced significantly higher complication rates and prolonged ICU stays following liver resection, even when undergoing minor procedures [36]. Similarly, Tanaka et al., using the KCL in a cohort of elderly Japanese patients, found that frailty was associated with increased 90-day mortality and functional decline, despite comparable 30-day morbidity across frail and non-frail groups [21]. These findings underscore that frailty in older adults may not always manifest through early complications but can profoundly delay or impair recovery—highlighting the importance of functional status assessment over age-based metrics.

Frailty is also a critical determinant of outcomes in patients with HCC and CRLM. Across multiple studies, frailty has been shown to predict not only postoperative complications but also long-term survival. In a meta-analysis by Lunca et al., frailty was strongly associated with increased rates of PHLF, major complications, and early postoperative mortality among patients undergoing oncologic liver resections [29]. Specifically, Okada et al. reported a stark contrast in 5-year overall survival between frail (42.7%) and non-frail (77.2%) HCC patients. Interestingly, recurrence-free survival was similar between groups, but frail patients were less likely to undergo curative salvage treatments, likely due to reduced physiologic reserve [22]. These data suggest that frailty not only heightens initial surgical risk but may also restrict access to repeat interventions following recurrence, thereby worsening long-term outcomes.

The extent of liver resection also modulates the clinical consequences of frailty. In patients undergoing major hepatectomy, a case–control analysis of 230 patients with liver malignancies spanning minor and major resections found that frailty significantly increased the risk of PHLF, ICU admission, and prolonged hospitalization [35]. In contrast, among those undergoing minor resections, frailty was not significantly associated with increased morbidity or mortality [38]. These findings support the strategic use of parenchymal-sparing techniques in frail patients to mitigate risk. Furthermore, Maegawa et al. demonstrated that laparoscopic liver resections were associated with fewer complications in frail individuals compared to open procedures [34]. This supports the use of minimally invasive surgery as a valuable approach in high-risk populations to reduce operative stress and enhance recovery. Taken together, these subgroup analyses reinforce that the impact of frailty is highly context dependent. Elderly and oncologic patients are particularly susceptible to poor outcomes when frailty is present, especially in the setting of major hepatic resections. However, when limited resections (parenchymal-sparing strategies) or minimally invasive techniques are employed, outcomes in frail patients may be acceptable and risks more manageable.

These insights emphasize the importance of tailoring operative strategies to patient physiology rather than relying solely on tumor burden or chronological age. Incorporating preoperative frailty assessment into routine clinical practice enables more precise surgical planning, ultimately optimizing outcomes across diverse patient populations.

### 7.2. How Frailty Influences Surgical Decision-Making and Patient Selection in Liver Resection

Frailty has become a key consideration in the preoperative selection of candidates for liver resection and in the tailoring of surgical strategies. Unlike chronological age, frailty offers a more accurate reflection of physiological reserve and has consistently demonstrated superior predictive value for adverse postoperative outcomes. In a multicenter analysis, Chen et al. found that the mFI was an independent predictor of both overall and major postoperative morbidity, while age alone lacked significant predictive value [32]. Similarly, Okabe et al. showed that frailty assessed by the CFS was significantly associated with severe complications and prolonged hospital stays after hepatectomy, whereas age did not reach statistical significance [19]. Frailty also plays a critical role in operative planning, particularly in determining the extent of liver resection. Frail patients undergoing major hepatectomy (defined as resection of ≥3 liver segments) face significantly higher risks of complications such as PHLF, ICU admission, and early postoperative mortality [35]. As a result, many centers now favor parenchymal-sparing approaches in frail patients, reserving extensive resections for those with greater physiological reserve. When technically feasible, minimally invasive techniques may further reduce operative risk. For example, Maegawa et al. reported that laparoscopic liver resection in frail individuals was associated with fewer complications compared to open surgery [34].

Beyond surgical planning, frailty also influences broader treatment decisions. Within multidisciplinary tumor boards, the presence of frailty often prompts consideration of non-surgical alternatives such as stereotactic body radiation therapy, transarterial chemoembolization, or systemic chemotherapy. For borderline surgical candidates, frailty screening can support triage into prehabilitation programs focused on improving nutritional status, reducing sarcopenia, and enhancing cardiopulmonary function. These interventions may increase surgical tolerance and, in some cases, allow previously ineligible patients to become operable [21].

Importantly, frailty assessment also informs shared decision-making. Patients and their caregivers should be counseled on the heightened risks associated with frailty, including an increased likelihood of serious complications, prolonged hospitalization, postoperative functional decline, and discharge to non-home settings [23]. Tools like the mFI, particularly when combined with liver-specific indices such as the ALBI grade, can enhance the accuracy of morbidity and mortality risk prediction [34], serving as valuable adjuncts in preoperative counseling.

A key priority for future research is the standardization of frailty assessment in hepatobiliary surgery, as the current diversity of tools and definitions limits comparability, weakens meta-analytic evidence, and impedes integration into clinical workflows. Building on existing data, a dual-pronged strategy appears most pragmatic. For clinical application, the Clinical Frailty Scale (CFS) stands out as a simple, validated, and intuitive bedside instrument that can be rapidly administered during surgical consultations. It captures a clinician’s holistic impression of a patient’s functional reserve and vulnerability, correlates strongly with postoperative morbidity and mortality, and can easily be embedded into preoperative screening, ERAS, and prehabilitation pathways. For research and registry-based analyses, the modified Frailty Index (mFI) remains the most practical choice, as it relies on routinely collected comorbidity and functional data, enabling reproducibility and large-scale risk adjustment across institutions and populations. Regionally, instruments such as the Kihon Checklist (KCL), widely used in Japan, add valuable granularity by integrating social and cognitive domains, making them particularly relevant in aging societies. Future multicenter collaborations should therefore adopt a core frailty dataset that includes one standardized clinical tool (e.g., CFS) and one database-compatible measure (e.g., mFI), together with uniform outcome reporting standards. Such harmonization would enhance cross-study comparability, refine perioperative risk stratification, and ultimately promote the integration of frailty assessment into personalized surgical planning, prehabilitation design, and long-term oncologic decision-making.

In summary, frailty is more than a prognostic indicator, it is a central determinant of the entire perioperative process. From assessing surgical candidacy and customizing procedural approaches to guiding treatment decisions and risk communication, frailty integration enhances the precision of liver surgery. It enables truly individualized care by aligning therapeutic strategies with patient resilience, physiological reserve, and personal goals.

## 8. Conclusions

Frailty represents a robust, multidimensional predictor of adverse outcomes in patients undergoing liver surgery. By reflecting overall physiological reserve and vulnerability, frailty assessment provides critical information that complements traditional liver-specific risk scores. Incorporating routine preoperative frailty evaluation, through validated and practical tools, should be regarded as a standard component of hepatobiliary surgical planning. To further validate its prognostic utility and optimize patient selection and perioperative management, well-designed prospective, multicenter studies are needed.

## Figures and Tables

**Figure 1 medsci-13-00253-f001:**
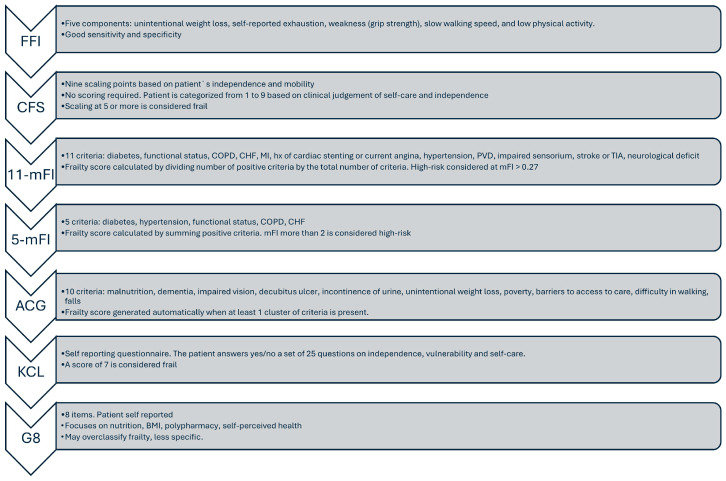
Overview of used frailty scales in liver surgery research. FFI, Fried frailty scale; CFS, clinical frailty scale; mFI, modified frailty index; ACG, Adjusted Clinical Group Johns Hopkins; KCL, Kihon checklist; G8, geriatric-8; COPD, chronic obstructive pulmonary disease; CHF, congestive heart failure; MI, myocardial infarction; PVD, peripheral vascular disease; TIA, transient ischemic attack.

**Table 1 medsci-13-00253-t001:** Summary of commonly used frailty assessment tools in liver surgery studies.

Tool	Type	Assessment Method	Components	Strengths	Limitations
**Fried frailty scale (FFI)**	Phenotypic model of frailty/Focuses primarily on physical frailty	Clinician-administered or research-based tool	Five observable criteria	Well-validated and widely used; Simple and objective; Physical focus	Only assesses physical aspects; self-reported components; less suitable for very ill, cognitively impaired
**Modified Frailty Index (mFI)**	Claims-based/Clinical	Derived from NSQIP variables	5- or 11-item version based on comorbidities (e.g., diabetes, CHF, COPD)	Easy to calculate from registry data; widely validated	Comorbidity-heavy; limited functional or cognitive assessment
**Clinical Frailty Scale (CFS)**	Clinical judgment	1–9 scale based on clinical impression	Assesses physical fitness, function, and independence	Quick and intuitive; validated in elderly populations	Subjective; inter-observer variability
**Kihon Checklist (KCL)**	Patient questionnaire	25 yes/no items	Covers nutrition, social, cognitive, and physical domains	Comprehensive multidimensional assessment	Lengthy; requires patient cooperation
Adjusted Clinical Groups **(ACG) Frailty Indicator**	Claims-based	Based on ICD-10 diagnostic clusters	Flags frailty from diagnoses like malnutrition, falls, dementia, etc.	Works in administrative datasets; scalable for large cohorts	Binary; lacks granularity and functional assessment
**Geriatric-8 (G8)**	Screening tool	8 items; mostly patient-reported	Focuses on nutrition, BMI, polypharmacy, self-perceived health	Good sensitivity for frailty screening in cancer patients	May overclassify frailty; less specific

**Key:** CHF: congestive heart failure; COPD: chronic obstructive pulmonary disease; NSQIP: National Surgical Quality Improvement Program; BMI: body mass index.

**Table 2 medsci-13-00253-t002:** Summary of Key Studies on Frailty and Short-Term Postoperative Outcomes after Liver Resection.

Study (Year)	Frailty Measure	Patient Cohort	Key Short-Term Outcomes (Complications, Mortality, LOS)
**Lv et al., 2024 [28]**	Meta-analysis (13 studies)	n = 84,096 (23,964 frail)	Frail patients had significantly increased risk of overall (RR~1.7) and major complications (RR~2.7). Mortality was substantially higher: 30-day mortality was 4.6× higher (RR 4.60), and 90-day mortality 2.5× higher (RR 2.52) in frail patients. LOS was prolonged by an average of 3.7 days.
**Lunca et al., 2024 [29]**	Meta-analysis (10 studies)	n = 71,102 (17,167 frail)	Frailty was linked to significantly higher morbidity, increased rate of major complications, and higher incidence of PHLF (all *p* < 0.001). Perioperative mortality and readmission rates were also significantly higher among frail patients.
**Zhang et al., 2025 [30]**	Not specified	n = 38,157 (35% frail)	Frail patients had increased odds of major complications (OR 4.01), and higher 30- and 90-day mortality risk (HRs 7.03 and 4.59, respectively), although these findings were not statistically significant.
**Louwers et al., 2016 [31]**	11-item mFI (≥1 = frail)	n = 10,300 (NSQIP)	Higher frailty scores were associated with increased rates of Clavien IV complications, 30-day mortality, and prolonged LOS. The relationship remained consistent across various types of hepatectomies.
**Chen et al., 2018 [32]**	5-item mFI (≥2 = frail)	n = 1928 (liver + colorectal resections)	Frail patients experienced significantly more overall and severe complications, longer LOS, and higher 30-day mortality (5.3% vs. 1.2%, *p* < 0.01). On multivariate analysis, frailty was an independent predictor of morbidity, whereas age was not.
**Shahrestani et al., 2023 [23]**	JHACG Frailty Indicator	n = 1515 (NRD)	Frailty was associated with increased inpatient complications (e.g., infections, DVT, UTI), higher in-hospital mortality, and prolonged LOS. Frail patients were more frequently discharged to nursing/rehabilitation facilities. Including frailty in prediction models improved outcome prediction over age alone.
**Maegawa et al., 2022 [34]**	5-item mFI (≥1 = frail)	n = 24,150 [NSQIP 2014–19]	Frailty was linked to higher rates of major complications, 30-day mortality, and PHLF. The mFI improved predictive accuracy when added to the ALBI score. Laparoscopic surgery was associated with better outcomes than open surgery in frail patients.
**Osei-Bordom et al., 2022 [37]**	mFI	n = 1826 (34.7% frail)	Frail patients had significantly higher 90-day mortality (6.6% vs. 2.9%) and postoperative complication rates (36.3% vs. 26.1%). LOS was longer for frail patients undergoing open surgery compared to laparoscopic, with similar trends observed in non-frail patients.
**Tanaka et al., 2018 [21]**	KCL (≥8 = frail)	n = 217 (≥70 years, multicenter)	Although overall complication rates were comparable between groups, frail patients had higher 90-day mortality (4.8% vs. 0%). Frailty independently predicted age-related adverse outcomes such as cardiopulmonary complications and functional decline.
**McKechnie et al., 2021 [38]**	mFI (≥0.27 = frail)	n = 409 (Canada, mixed tumors)	Frail patients had significantly more postoperative complications (79% vs. 46%) including major (50% vs. 13%) and minor (69% vs. 42%) events, longer median LOS (9.5 vs. 5 days), and higher 90-day mortality (12% vs. 3.4%). Frailty independently predicted major complications.

**Key:** RR: relative risk; OR: odds ratio; HR: hazards ratio; LOS: length of stay; PHLF: post-hepatectomy liver failure; JHACG: Johns Hopkins Adjusted Clinical Groups; mFI: modified frailty index; KCL: Kihon Checklist; DVT: deep vein thrombosis; UTI: urinary tract infection; ALBI: Albumin-Bilirubin; NSQIP: National Surgical Quality Improvement Program; NRD: National Readmissions Database.

**Table 3 medsci-13-00253-t003:** Summary of Key Studies on Frailty and Long-Term Survival after Liver Resection.

Study (Year)	Frailty Measure	Patient Cohort & Follow-Up	Long-Term Survival Findings
**Lv et al., 2024 [28]**	Meta-analysis	Various (meta-analysis of 13 studies), ~5-year outcomes (pooled)	Frail patients demonstrated significantly poorer long-term survival across pooled studies, with a nearly threefold increased hazard of death (HR 2.89, 95% CI: 1.84–4.53).
**Okada et al., 2024 [22]**	KCL (frail = KCL ≥ 8)	n = 81, ≥65 y with HCC (prospective; median 36 mo follow-up)	Five-year overall survival was markedly lower in frail patients (42.7%) compared to non-frail (77.2%) (*p* = 0.005). Frailty independently predicted worse OS on multivariate analysis. While DFS did not differ significantly, frail patients experienced more extrahepatic recurrences and underwent fewer salvage treatments.
**Yamada et al., 2021 [36]**	CFS (frail = CFS ≥ 4)	n = 92, >75 y with HCC (mean follow-up 2.6 years)	Frail patients had significantly reduced cancer-specific 3-year survival (72.0% vs. 94.3%, *p* < 0.01) and OS (*p* < 0.01). DFS was also significantly worse in the frailty group (*p* = 0.01). Multivariate analysis identified frailty as the only independent prognostic factor. Frail patients had a higher rate of extrahepatic recurrence (50% vs. 4.8%) and were significantly less likely to receive treatment for recurrence (50% vs. 95.2%).
**Tokuda et al., 2021 [20]**	CFS (frail = CFS ≥ 4)	n = 87, median age 78 (CRLM; mean follow-up 46.2 months)	Three-year OS and CSS were significantly lower in frail patients (OS: 63.9% vs. 89.1%; CSS: 69.3% vs. 91.0%; both *p* < 0.01). Frailty was the only independent predictor of worse survival (*p* = 0.0477). No significant difference in recurrence rates
**Hosoda et al., 2022 [40]**	CFS (score 1–2 = non-frail, 3–9 = frail)	n = 87, mean age 71 (perihilar cholangiocarcinoma)	Five-year OS was significantly lower in frail patients (10.2%) compared to non-frail (41.8%) (*p* = 0.01), with a similar trend for disease-specific survival. This survival gap was more pronounced in early-stage disease (stage 0–II: 44.5% vs. 13.0%; *p* = 0.02), whereas outcomes in advanced-stage patients (stage III/IV) showed no significant difference (OS: 40.0% vs. 0%; *p* = 0.46). On multivariate analysis, a CFS score of 3–9 remained an independent predictor of OS (HR 2.31, 95% CI: 1.14–4.87; *p* = 0.02).

**Key:** KCL: Kihon Checklist; CFS: Clinical Frailty Scale; HCC: hepatocellular carcinoma; CRLM: colorectal liver metastases; HR: hazards ratio; CI: confidence interval; OS: overall survival; DFS: disease free survival; CSS: cancer-specific survival.

## Data Availability

No new data was created or analyzed in this study as this review synthesized existing literature.

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
