# Peer review of "Should We Fear the Frail? A Review on the Impact of Frailty on Liver Surgery"

_medsci, 2025, doi:10.3390/medsci13040253_

Round 1
Reviewer 1 Report
Comments and Suggestions for Authors
To the Authors,
the manuscript presents a comprehensive and timely review of the impact of frailty on outcomes following liver resection. The topic is of paramount importance as the demographic shift towards an older surgical population necessitates more refined risk assessment tools beyond chronological age and standard comorbidity indices. The authors successfully synthesize a substantial body of recent literature, including meta-analyses and large database studies, to build a compelling case for the independent predictive value of frailty. The structure is logical, progressing from definitions and assessment tools to prevalence, short- and long-term outcomes, and finally clinical implications. The writing is generally clear and the conclusions are well-supported by the cited evidence.
This review makes a valuable contribution to the field and is worthy of publication following some revisions that would enhance its methodological clarity, critical analysis, and clinical applicability.
Strengths: Comprehensive Scope: The review covers all critical aspects of the topic, including a clear explanation of different frailty models (phenotype vs. deficit accumulation) and a practical overview of the most relevant assessment tools (mFI, CFS, KCL, etc.), complete with a helpful table summarizing their strengths and limitations. Current and Relevant Synthesis: The authors have incorporated very recent, high-impact meta-analyses (e.g., Lv et al. 2024, Lunca et al. 2024) and large-scale cohort studies, making the review state-of-the-art. The data presented is robust and convincing. Clinical Relevance: The sections on "Independent Predictive Value" and "How Frailty Influences Surgical Decision-Making" are particularly strong. They successfully translate the epidemiological findings into actionable clinical advice, emphasizing the importance of prehabilitation, parenchymal-sparing resections, and minimally invasive approaches for frail patients. Subgroup Analysis: The dedicated section on specific patient populations (elderly, HCC, CRLM) adds significant depth, acknowledging that the impact of frailty is not uniform and must be contextualized.
Weaknesses and Suggestions: Methodology of the Review: the manuscript would be significantly strengthened by a brief description of the literature search strategy. While it is a narrative review, stating the databases searched (e.g., PubMed, Embase), search terms, and the time frame covered would enhance its reproducibility and scholarly rigor. A PRISMA-style flow diagram is not mandatory for a narrative review, but a sentence or two on the methodology is expected. Critical Analysis of Conflicting Evidence: the authors correctly note that some studies (e.g., Tanaka et al.) found no difference in 30-day mortality or overall complications. However, the discussion of why these discrepancies might exist is somewhat superficial. I recommend a more critical analysis of these conflicting results. Potential explanations could include: Selection Bias: In single-center or prospective studies, surgeons may have intuitively avoided high-risk operations in frail patients, thus attenuating the observed effect. Type of Resection: The authors mention this briefly, but it deserves emphasis. The effect of frailty is likely magnified in major hepatectomies versus minor/wedge resections. Regional Differences in Perioperative Care: Variations in prehabilitation programs, enhanced recovery after surgery (ERAS) protocols, and social support systems between Japan, Europe, and the US could influence outcomes. Nuance in Long-Term Outcomes: The finding that frailty impacts overall survival but not necessarily recurrence-free survival is crucial. The authors suggest this is due to a reduced ability to receive salvage therapy. This point should be further developed. It underscores that frailty is not just a predictor of surgical risk but a marker of overall physiological decline that affects a patient's entire cancer journey. This has profound implications for informed consent and goals-of-care discussions. Standardization and Future Directions: The conclusion calls for standardization in future trials but is vague on how to achieve this. The authors should provide a more specific recommendation. For instance, they could propose which 1-2 tools (e.g., CFS for clinical use, mFI for database research) they believe are most pragmatic for widespread adoption in hepatobiliary surgery, based on their extensive review.
Minor Corrections. Introduction, Paragraph 2: The statement "other nonsurgical alternatives such as thermos-ablation or chemoembolization are reserved as a second line therapy" is an oversimplification. For very early-stage HCC in unsuitable surgical candidates, ablation is a first-line curative option per major guidelines (e.g., AASLD, EASL). Please rephrase to reflect this nuance. Section 4: When discussing the Osei-Bordom et al. study, the text references "Lunca et al., 2024 (29)" in parentheses. This appears to be a citation error; it should likely reference Osei-Bordom et al. directly. Please check all in-text citations for accuracy.
Conclusion This is a well-constructed, informative, and clinically valuable review article. It effectively consolidates the current evidence demonstrating that frailty is a powerful, independent predictor of adverse outcomes in liver surgery. The requested revisions aim to bolster the methodological transparency and critical depth of an already strong manuscript. I am confident that after these revisions, this article will be a significant resource for hepatobiliary surgeons, oncologists, and all clinicians involved in the care of patients requiring liver resection.
Comments on the Quality of English LanguageLanguage and Flow: The manuscript is well-written but would benefit from a final proofread by a native English speaker for minor grammatical polish (e.g., "patient’s" should be "patient's" or "patients'" depending on context).
Author Response
Dear reviewer,
Thank you for giving us the opportunity to submit a revised manuscript. We appreciate the time and effort that you dedicated to our study, and we are grateful for the interesting comments and valuable suggestions you have made to the paper.
In this revised manuscript we have incorporated most of the comments and highlighted the changes to the text in red. Please find bellow our point-by-point replies and changes to your suggestions.
Comment 1: the manuscript would be significantly strengthened by a brief description of the literature search strategy. While it is a narrative review, stating the databases searched (e.g., PubMed, Embase), search terms, and the time frame covered would enhance its reproducibility and scholarly rigor. A PRISMA-style flow diagram is not mandatory for a narrative review, but a sentence or two on the methodology is expected.
Answer: Thank you for this suggestion. Indeed, this being a narrative review we did not report the search strategy, however this was done thoroughly and we have now added a short paragraph in the Methods section describing the databases searched (PubMed, Scopus, Embase), the search terms (“frailty,” “liver resection,” “hepatectomy,” “outcomes”), and the period covered (earliest date to 2025). A PRISMA flow diagram was not included because this is a narrative review, but reproducibility is now improved.
Comment 2: the authors correctly note that some studies (e.g., Tanaka et al.) found no difference in 30-day mortality or overall complications. However, the discussion of why these discrepancies might exist is somewhat superficial. I recommend a more critical analysis of these conflicting results. Potential explanations could include: Selection Bias: In single-center or prospective studies, surgeons may have intuitively avoided high-risk operations in frail patients, thus attenuating the observed effect. Type of Resection: The authors mention this briefly, but it deserves emphasis. The effect of frailty is likely magnified in major hepatectomies versus minor/wedge resections.
Answer: Thank you for your comment. We have expanded the Discussion to analyse this aspect (Lines 253-268)
Comment 3: Variations in prehabilitation programs, enhanced recovery after surgery (ERAS) protocols, and social support systems between Japan, Europe, and the US could influence outcomes.
Answer: Thank you for your suggestion. In the revised version we have added a new paragraph commenting on the above (Lines 385-408)
Comment 4: The finding that frailty impacts overall survival but not necessarily recurrence-free survival is crucial. The authors suggest this is due to a reduced ability to receive salvage therapy. This point should be further developed. It underscores that frailty is not just a predictor of surgical risk but a marker of overall physiological decline that affects a patient's entire cancer journey. This has profound implications for informed consent and goals-of-care discussions.
Answer: Thank you. We have added a new paragraph discussing the role of frailty on long term outcomes (Lines 329-344)
Comment 5: The conclusion calls for standardization in future trials but is vague on how to achieve this. The authors should provide a more specific recommendation. For instance, they could propose which 1-2 tools (e.g., CFS for clinical use, mFI for database research) they believe are most pragmatic for widespread adoption in hepatobiliary surgery, based on their extensive review.
Answer: Thank you for this suggestion. This has been addressed with a new paragraph emphasizing the specific use of each tool with clinical recommendations (Lines 501-520).
Comment 6: The statement "other nonsurgical alternatives such as thermos-ablation or chemoembolization are reserved as a second line therapy" is an oversimplification. For very early-stage HCC in unsuitable surgical candidates, ablation is a first-line curative option per major guidelines (e.g., AASLD, EASL). Please rephrase to reflect this nuance.
Answer: Thank you. We have addressed this.
Comment 7: When discussing the Osei-Bordom et al. study, the text references "Lunca et al., 2024 (29)" in parentheses. This appears to be a citation error; it should likely reference Osei-Bordom et al. directly. Please check all in-text citations for accuracy.
Answer: This has been amended to.
Reviewer 2 Report
Comments and Suggestions for Authors
This is a comprehensive review of the frailty assessment tools for assessing the survival risk for patients undergoing major surgery with a major focus on liver resection. This review is very similar to your previous publication in the Annals of Surgical Oncology published in 2024. It is reference #29 in the current manuscript. Many of the same references are in both manuscripts.
The conclusions of the previous publication were "Frailty seems to be a solid predictive risk factor of morbidity and mortality after liver surgery and should be considered a selection criterion for liver surgery in at-risk patients"
This latest manuscript goes in to more detail on the published frailty scales and examines the published outcomes and concludes "Fraility is a strong and independent predictor of poor outcomes after liver resection." It backs up your previous conclusion with more data.
This manuscript supports the conclusions of the previous manuscript. My concern is that many of the clinical trials referenced in this manuscript were also referenced in the previous manuscript. There is a repetition but you have included extra studies that support your findings in this manuscript, so I believe that this added support emphasizes your previous findings.
Author Response
Dear reviewer,
Thank you for giving us the opportunity to submit a revised manuscript. We appreciate the time and effort that you dedicated to our study, and we are grateful for the interesting comments and valuable suggestions you have made to the paper.
In this revised manuscript we have incorporated most of the comments and highlighted the changes to the text in red. Please find bellow our point-by-point replies and changes to your suggestions.
Comment 1: This review is very similar to your previous publication in the Annals of Surgical Oncology published in 2024. It is reference #29 in the current manuscript. Many of the same references are in both manuscripts.
Answer: Thank you for this observation. The current manuscript builds upon the prior review by focusing specifically on frailty tools, scale validation, and cross-study outcome synthesis, thereby extending the scope rather than repeating the earlier analysis.
Comment 2: The conclusions of the previous publication were "Frailty seems to be a solid predictive risk factor of morbidity and mortality after liver surgery and should be considered a selection criterion for liver surgery in at-risk patients. This latest manuscript goes in to more detail on the published frailty scales and examines the published outcomes and concludes "Fraility is a strong and independent predictor of poor outcomes after liver resection." It backs up your previous conclusion with more data.
Answer: We thank the reviewer. The present review supports and refines this conclusion by integrating newer studies and by contextualizing frailty within prehabilitation and ERAS frameworks. Also, herein we provided a more in-depth comparison between frailty assessment tools and provide a better recommendation on which to use in different clinical scenarios.
Comment 4: This manuscript supports the conclusions of the previous manuscript. My concern is that many of the clinical trials referenced in this manuscript were also referenced in the previous manuscript. There is a repetition but you have included extra studies that support your findings in this manuscript, so I believe that this added support emphasizes your previous findings.
Answer: Thank you for this comment. Indeed, many of the studies were also included in the previous meta-analysis, however here we added newer papers which bring new insight regarding the role of frailty, which couldn`t have been included in the meta-analysis as we were restricted by the meta-analytical format.